# Diabetic Neuropathic Cachexia: A Clinical Case and Review of Literature

**DOI:** 10.3390/life12050680

**Published:** 2022-05-04

**Authors:** Alessio Bellelli, Daniele Santi, Manuela Simoni, Carla Greco

**Affiliations:** 1Unit of Endocrinology, Department of Biomedical, Metabolic and Neural Sciences, University of Modena and Reggio Emilia, 42121 Modena, Italy; alessio.bellelli@gmail.com (A.B.); daniele.santi@unimore.it (D.S.); manuela.simoni@unimore.it (M.S.); 2Unit of Endocrinology, Department of Medical Specialties, Azienda Ospedaliero-Universitaria di Modena, Ospedale Civile di Baggiovara, 41125 Modena, Italy

**Keywords:** diabetic neuropathy, neuropathic pain, weight loss, cachexia, diabetes mellitus

## Abstract

A 46-year-old man was admitted to the surgical department because of abdominal pain and anemia, with the radiological finding of a perforated duodenal ulcer, and underwent laparoscopic surgical treatment. Type 2 diabetes mellitus (T2DM) had been diagnosed 5 years earlier and treated with diet. At clinical investigation, the patient was depressed and anorexic; moreover, he complained of lower extremity weakness and bilateral feet pain, burning in nature and accompanied by allodynia. This painful sensation had been preceded by an 8-month history of fatigue and anorexia with profound weight loss of 35 kg. After clinical evaluation and a nerve conduction study, diagnosis of diabetic cachectic neuropathy was made based on the rapid onset of severe neuropathic pain in the context of diabetic neuropathy, marked weight loss, and depressed mood. The therapy with pregabalin and duloxetine had scarce effect and was gradually discontinued. The patient, however, obtained progressive relief and amelioration of neuropathic lower-limb pain concomitant with weight gain. This clinical trend also confirmed the diagnosis of this rare form of diabetic neuropathy. A few cases of diabetic neuropathic cachexia have been reported in the literature and are briefly reviewed here.

## 1. Introduction

The term “diabetic neuropathic cachexia” was first suggested in 1974 to describe patients with diabetes mellitus (DM) complicated by bilateral distal symmetrical peripheral neuropathy (DSPN) with severe pain [1]. The first six male patients showed severe emotional disturbances, anorexia with profound weight loss, and uniformly spontaneous recovery of pain symptom in about one year upon restoration of previous weight levels [1]. The cachectic appearance was described in all these patients and the term “diabetic neuropathic cachexia” was suggested [1]. Hitherto, few additional cases have been reported in the literature, in which profound weight loss and severe pain in the context of diabetic neuropathies are the main manifestations.

DSPN represents the most clinically relevant manifestation of typical forms of diabetic neuropathy (DN) [2]. DSPN presents as a slowly progressive, symmetrical, primarily sensory deficit in a length-dependent manner, with symptoms starting in the feet and then spreading more proximally, evoking the classic stocking-glove distribution [2]. In particular, the painful diabetic polyneuropathy (PDPN) is the form of DSPN with chronic neuropathic pain arising as a direct consequence of lesions or dysfunction in the peripheral somatosensory nervous system [2,3]. Less common clinical conditions in the DN context include mononeuritis multiplex, radiculoplexus neuropathy, thoracic radiculopathy, and non-diabetic neuropathies common in DM such as pressure palsies, chronic inflammatory demyelinating polyneuropathy, and acute painful treatment-induced small-fiber neuropathies [4].

In this complex setting, diabetic neuropathic cachexia should be considered as an even rarer clinical entity in DM with peculiar clinical and prognostic characteristics. We herein report a recent case of diabetic neuropathic cachexia and compare it to previous cases in the literature.

## 2. Case Presentation

M. M., a 46-year-old man, was admitted to the surgical department of Modena Hospital (Italy) in January 2021 for abdominal pain and anemia with radiological finding of a perforated duodenal ulcer, treated with emergency laparoscopic surgery. Furthermore, gastric ulcer class III of Forrest classification [5] with positive antigen test for Helicobacter pylori infection and peritonitis were found endoscopically.

At hospital admission, the patient showed type 2 DM (T2DM) diagnosed 5 years earlier, treated with diet alone and always in good control (HbA1c values up to 53 mmol/mol) in the previous period. Moreover, he presented a deflected mood, loss of interest in daily activities, anhedonia, loss of appetite, and sleep disturbances. Although a structured assessment of depression was not performed, these symptoms and signs confirmed a depressive mood disorder. In fact, the patient was depressed and anorexic; he reported voluntary reduction in nutrition intake in the previous months due to family issues. Moreover, he complained of lower extremity weakness and pain started five months before hospital admission and was characterized by bilateral feet pain, burning in nature accompanied by allodynia with high severity (score of 9/10 at Visual Analog Scale (VAS)). Additionally, electric shocks, pinpricks, pins and needles were reported in the same area of the lower limbs. The pain was persistent and more severe at night and disturbing the patient’s sleep. In addition, the patient reported to be functionally impaired in daily activities and could not go to work because of persistent pain and fatigue. This painful sensation was preceded by an 8-month history of fatigue and anorexia with profound weight loss of 35 kg. 

A surgically treated non-metastatic melanoma (2014) and a botulinum toxin treatment for hand hyperhidrosis (childhood–adolescence period) were reported at medical history collection. A past smoking history up to 20 years was reported, together with a previous and current low alcohol consumption. The patient reported taking non-steroidal anti-inflammatory drugs (NSAIDs) in the weeks prior to hospital admission for pain in the lower limbs. No other chronic drug therapy was prescribed prior to admission. Written informed consent was obtained from the patient for publication of this report.

## 3. Investigation

On physical examination, the patient showed signs of generalized loss of subcutaneous fat tissue and widespread muscle hypotrophy and seemed depressed. He was 184 cm tall and weighed 75 kg (body mass index (BMI): 22.15 m/kg^2^). The previous weight was 110 kg (BMI 32.49 kg/m^2^). Supine blood pressure was 110/70 mmHg and heart rate 75 beats per minute. 

As recommended [4], we performed a detailed neurological clinical evaluation for peripheral neuropathy. First, we evaluated the presence of DSPN symptoms by the Michigan Neuropathy Screening Instrument (MNSI)-questionnaire [6]. The MNSI-questionnaire documented subjective sensation of numb feet, burning pain, sensitive to touch, pinpricks, inability to distinguish temperature differences, weakness and worsening of symptoms at night (score 8/13, abnormal). A check of peripheral sensitivity perceptions was performed using the structured systems of clinical examination: Diabetic Neuropathy Index (DNI) from MNSI examination and Michigan Diabetic Neuropathy Score (MDNS) [6], the latter including the assessment of strength in the upper and lower limbs. At inspection, the feet showed no deformities, callus, ulcer, or signs of infection. On examination, deep tendon reflexes were intact in the upper limbs (biceps and triceps), reduced for patellar, and absent ankle jerks. There was diminished sensation to light touch on the sole of the left foot, otherwise normal on the contralateral side. Other sensory tests (such as 10 g monofilament perception, vibration perception thresholds, and pin prick in the big toes) were normal. No hyperalgesia and/or allodynia in the same area of reported pain were recorded. The strength study documented a slight deficit of extension of the big toe on the left, normal for the rest (score DNI 2/8, normal; score MDNS 7/46, abnormal). Furthermore, to identify the nature of reported pain, we utilized the questionnaire Douleur Neuropathique en 4 Questions (DN4), a screening tool for neuropathic pain consisting of interview questions (DN4-interview) and a brief physical test [7,8]. DN4 score was 5/10 and was indicative of neuropathic pain. The nerve conduction study (NCS) revealed slow sensory nerve conduction velocities and reduced amplitudes predominantly affecting the lower extremities consistent with a moderate axonal DSPN. The Doppler study of the arterial vascular system of the lower limbs was normal. Thus, a diagnosis of diabetic neuropathic cachexia was made based on the rapid onset of severe neuropathic pain in the DN context, marked weight loss and depressed mood, concomitantly with duodenal-gastric ulcer disease.

Further, the patient underwent an extensive screening workup. The chest and bilateral foot radiograph and abdominal ultrasound were unremarkable. The esophagogastroduodenoscopy showed gastric and duodenal ulcer outcomes. Serum laboratory studies including basic hematology, biochemistry, autoimmune screening, and inflammatory markers were compatible with the gastric-intestinal disorder encountered. In particular, macrocytic anemia (hemoglobin 9.6 g/dL and mean corpuscular volume 119.1 fL) was detected at hospital admission. Vitamin B12 (471 pg/mL) and folic acid (2.1 ng/mL) were detected within reference ranges, according to the supplemental therapy started a few weeks before hospitalization on the recommendation of the attending physician. In addition, vitamin D levels were severely deficient (5.9 ng/mL). Antiparietal cell antibodies (antibody anti-transglutaminase) and DQ2-DQ8 HLA typing were negative. Thyroid function test for the differential diagnosis of weight loss was normal. Severely high fasting blood glucose (319 mg/dL) was detected, most likely as stress hyperglycemia considering pathological condition and treatment. Furthermore, high inflammation marker (C-reactive protein 19.3 mg/dL) was observed, and the low albumin serum level (2.8 g/dL) confirmed the protein malnutrition detected at physical examination. The dosage of glycated hemoglobin (HbA1c) was 51 mmol/mol, in any case to be interpreted in light of recent anemia. C peptide was normal at 2.16 ng/mL. No evidence of other diabetic microvascular complications was found. Fundus oculi examination revealed no diabetic retinopathy and urinalysis showed albumin 13 mg/g of creatinine, otherwise normal. 

## 4. Treatment during Hospitalization

During the hospital stay, the patient was treated with nutrition support, initially parenteral and thereafter enteral. The artificial nutrition was performed together with insulin to optimize glycemic control, whereas insulin was not required during natural oral nutrition. Furthermore, adequate hydration and supplementation of vitamins (B group and D) were guaranteed. No complications were noted during the process of nutritional rehabilitation. 

For the neuropathic pain, medication was initially pregabalin at 75 mg twice daily (indication to titration up to 450 mg per day). This had little effect and duloxetine 60 mg daily was further introduced. Finally, in consideration of the finding of gastrointestinal ulcers, antibiotic eradication of Helicobacter pylori and surgical treatment were conducted.

## 5. Outcome and Follow-up

After hospital discharge, the patient’s distal pain significantly improved in a 6-month follow-up. Since pregabalin and duloxetine had scarce effect on neuropathic pain, they were gradually discontinued 5–6 weeks after discharge. The patient’s progressive relief and amelioration of neuropathic lower-limb pain was obtained concomitantly with weight gain. The follow-up examination performed six months after hospital discharge showed a well-controlled T2DM (HbA1c 47 mmol/mol) with metformin therapy, a body weight gain of 20 kg (body weight 95 kg; BMI 28 m/kg²), and a complete relief of neuropathic pain (MNSI score 1/13 and VAS 0/10). No more walking disturbance and weakness was detected. This overall condition led to depressed mood resolution.

## 6. Discussion

We reported a case of rare clinical form of DN in a middle-aged man with moderate diabetes control, called diabetic neuropathic cachexia. The patient presented with acute neuropathic pain at feet concomitant to profound and rapid weight loss and depressed mood. Sensory loss and motor signs were mild or absent, and reduced patellar and absent ankle reflexes were found. The improvement in pain concomitant with weight gain allowed to confirm the diagnosis of this rare kind of DN. 

Few cases of diabetic neuropathic cachexia have been reported so far. Using a MEDLINE search using diabetic neuropathic cachexia and acute painful neuropathy cachexia as key words, only 15 articles and 31 case reports of this DN rare variant were detected [1,9,10,11,12,13,14,15,16,17,18,19,20,21,22]. Table 1 summarizes the clinical aspects of cases described so far. Considering the clinical characteristics, weight loss, acute pain, and depressed mood are the main elements common to all cases described. Indeed, moderate to marked weight loss always occurs, ranging from 12 to 57% of the patient’s original weight. In our case, 35 Kg weight reduction was documented, corresponding to approximately 32% of the initial weight. Moreover, most of the reported cases concern males (23 of 31 subjects) between the fourth and sixth decades of life. The overall age at diagnosis ranges from 13 to 71 years with an average age of 48 years. The mean age at diagnosis is 51 years in males and 40 years in females. Thus, our case enters within this typical age distribution of diabetic neuropathic cachexia in males. Moreover, differences in type of diabetes and glycemic control have been observed among all cases available in the literature (Table 1). In particular, the condition of diabetic neuropathic cachexia does not seem to be linked to the severity of diabetes and to be monophasic typically over several months [23]. This peculiarity of this rare DN condition suggests that a potential mechanism different to glycemic control is involved in its pathogenesis.

Alongside weight loss, depression is often present in the literature. Accordingly, our patient showed depression and sleep disturbances at diagnosis, exacerbated by foot pain. No psychiatric history was previously known. The pain was bilateral at the feet, burning and accompanied by allodynia, electric shocks, pinpricks, and pins and needles. As known, neuropathic pain arises from some lesion or disease of the somatosensory system [3], due to different contributing mechanisms, such as aberrant ectopic activity in nociceptive nerves, peripheral and central sensitization, impaired inhibitory modulation, and pathological activation of microglia [24]. Clinically, spontaneous neuropathic pain with debilitating symptoms such as allodynia and hyperalgesia have a substantial negative impact on patients’ quality of life, psychiatric disorders, and occupational dysfunction. Indeed, a high prevalence of anxiety, depression, and sleep comorbidities exists among diabetic patients with chronic neuropathic pain [25]. In particular, PDPN has been observed as a greater determinant of depression among diabetic people enhancing depression severity more than other diabetes-related complications and comorbidities [26]. In our patient, depression preceded chronologically the neuropathic pain onset, later contributing to the maintenance and exacerbation of pain and of sleep disorders. Depression, meanwhile, could also have adversely affected pain behaviors ranging from symptomology to treatment response. Pain and depression are important interconnected comorbidities that could independently induce long-term plasticity in the central nervous system [27]. In this interplay, pain can cause depression, and depression can worsen pain behaviors as clearly indicated from clinical and preclinical studies [27]. Therefore, depression can explain poor nutrition of the patient and the consequent weight loss as well as contributing to the maintenance of pain.

Sexual dysfunctions such as erectile disturbance were frequent features in the collected cases of diabetic neuropathic cachexia. Therefore, it is likely that the combination of DN, depression, and acute pain can significantly contribute to the erectile deficit from a psychological point of view. Unfortunately, this aspect was not evaluated in our patient.

Considering the therapeutic aspect, the use of neuropathic-pain-specific drugs alone or in combination with common analgesics was described in previous clinical cases (Table 1). Gabapentinoids, selective serotonin reuptake inhibitors (SSRIs), tricyclic antidepressants (TCAs), sodium channel blockers, dopamine receptor antagonists (DRAs) alone or in addition to non-steroidal anti-inflammatory drugs (NSAIDs) or common analgesics, have been previously used to treat neuropathic pain with varying effectiveness. Narcotic drugs and benzodiazepines are used for sleep disorders. However, the effectiveness of drugs against neuropathic pain in diabetic neuropathic cachexia is still unclear. Our patient was treated initially with pregabalin at 75 mg twice daily and then, given the scarce effect, with serotonin-norepinephrine reuptake inhibitor (SNRI) duloxetine 60 mg daily. Both pregabalin and duloxetine showed poor efficacy on pain and they were gradually discontinued. Progressive relief and amelioration of neuropathic lower-limb pain was obtained six months later, concomitantly with weight gain. With this in mind, it could be speculated that the appropriate therapeutic choice in diabetic neuropathic cachexia is re-alimentation and recovery of weight. Indeed, adequate hydration and nutritional support were guaranteed in our patient together with supplementation of vitamins. Within the therapeutic strategy aiming at correcting malnutrition, an important role is played by vitamin B12 supplementation. Indeed, vitamin B12 deficiency is quite common in patients with T2DM and may cause or accelerate progression of neurological disorders, such as peripheral, autonomic, and painful neuropathy [28,29]. In this regard, improvement in neurological parameters of peripheral sensitive and autonomic functions with oral methylcobalamin for 12 months was recently shown [30]. In our case, the supplementation of group B vitamin (B6 + B9 + B12) was promptly performed, together with protein support. Since B12 integration continued when natural nutrition was resumed and after hospital discharge, a role of B12 in the clinical course of our patient cannot be excluded.

The rapid onset of the triad depression, profound weight loss, and acute neuropathic pain followed by an improvement in symptoms within 6–12 months, i.e., a transient symptomatology, represents the main characteristic of diabetic neuropathic cachexia. In this context, the most intriguing feature is the propensity of symptoms to spontaneously resolve. Indeed, together with improving depression, anorexia decreased, leading to weight gain. Thereafter, the patient recovered the ability to walk without trouble, resuming his normal lifestyle. Moreover, he simultaneously showed a clear improvement in neuropathic pain even without specific pharmacological treatments. Therefore, the interplay between weight recovery and pain relief appears difficult to explore. Considering the clinical history of our patient, a role of the adipose tissue on neuropathic pain is likely. The rapid weight loss could have led to an excessive reduction in muscle mass and sarcopenia. Indeed, the patient showed signs of generalized widespread muscle hypotrophy on physical examination. Moreover, low albumin serum level confirmed protein malnutrition. In fact, in our case, as in others described, the term ‘cachexia’ mainly refers to the physical appearance of the subjects and the condition of protein–energy malnutrition resulting in sarcopenia, rather than weight and/or BMI value. Sarcopenia leads to increase in the ratio of adipose tissue to muscle mass, even if weight and/or BMI is in the normal range. Some studies underline the link between sarcopenia and chronic diabetic neuropathy, especially in patients with sarcopenic obesity [31]. This disproportionate increase in adiposity can chronically sustain a systemic low grade inflammatory state, which can impact a number of secondary conditions including neuropathic pain. Indeed, adipocytes are a major neuroendocrine organ that continually and systemically releases a number of proinflammatory cytokines, including tumor necrosis factor α (TNFα), interleukin-1b (IL-1b), interleukin-6 (IL-6), monocyte chemoattractant protein-1 (MCP-1), and nuclear factor κB (NFκB). These cytokines sensitize nociceptive neurons by increasing the expression and phosphorylation of voltage-gated sodium channels (NaV) and transient receptor potential vanilloid (TRPV) [32] and can therefore reinforce hyperresponsivity within the nociceptive system as is the case with neuropathic pain. Therefore, weight regain with improvement in muscle mass and walking ability may have contributed to pain remission, along with the improvement in depression.

Finally, the rapid onset of neuropathic pain and its characteristics could lead to suspicion of another form of diabetic painful neuropathy, called treatment-induced neuropathy of diabetes (TIND). TIND is an iatrogenic painful neuropathy that may be triggered by a rapid decline in blood glucose levels following the use of insulin, other anti-diabetic drugs, or even diet [33]. Usually, a decrease in HbA1c of more than 3% points in 3 months in individuals with chronic hyperglycemia increases the risk of developing TIND [33]. In our case, the HbA1c value at admission and previous stable control of diabetes led to the exploration of different diagnostic hypothesis. In addition, the extensive screening workup during hospitalization allowed to exclude paraneoplastic forms of weight loss and pain. Moreover, another type of diabetic peripheral neuropathy called diabetic amyotrophy could be considered in the differential diagnosis as well. Weight loss is also common in this type of neuropathy, with acute to subacute onset of severe lower extremity pain [34]. However, asymmetrical involvement at lower extremity in the thigh or the leg and incomplete pain recovery in diabetic amyotrophy as opposed to presentation in diabetic neuropathic cachexia are the main differences [33]. These features further point towards diabetic neuropathic cachexia as the final diagnosis for our patient.

## 7. Conclusions

In conclusion, diabetic neuropathic cachexia is a distinct entity in the DN context characterized by bilateral feet neuropathic pain with dramatic weight loss in diabetic subjects. The pathogenic mechanism is still unknown. Hitherto, most patients were middle-aged type 2 diabetics on non-insulin antidiabetic therapy. Diabetic neuropathic cachexia has rarely been reported in patients with type 1 diabetes. No specific pharmacological therapy is useful and most patients recover spontaneously with negligible residual clinical manifestations. The effectiveness of drugs against neuropathic pain is unclear, whereas the use of antidepressants seems to be more rational. Adequate nutrition allows the recovery of normal function and could contribute to the remission of symptoms.

## Figures and Tables

**Table 1 life-12-00680-t001:** Clinical aspects of the previously published cases of diabetic neuropathic cachexia.

Case (Author, Year)	Gender	Age (Years) at Diagnosis	Type of Diabetes	Diabetes Duration (Years)	Antidiabetic Therapy at Diagnosis	Glycemic Control at Diagnosis	Weight Loss (%) at Diagnosis	Depression Mood at Diagnosis	Erectile Dysfunction at Diagnosis	Therapy of Pain	Time of Recovery of Pain	Time of Recovery of Weight
Case 1, Ellenberg, 1974 [1]	M	60	n.a.	0	acetohexamide	well-controlled	35	+	+	n.a.	12 months	12 months
Case 2, Ellenberg, 1974 [1]	M	51	n.a.	1	tolbutamide, insulin	poorly controlled	57	+	+	n.a.	n.a.	24 months
Case 3, Ellenberg, 1974 [1]	M	58	n.a.	0	Acetohexamide, insulin	poorly controlled	41	+	+	n.a.	n.a.	18 months
Case 4, Ellenberg, 1974 [1]	M	35	n.a.	n.a.	chlorpropramide	well-controlled	27	+	+	narcotic drugs	4 months	12 months
Case 5, Ellenberg, 1974 [1]	M	63	n.a.	0	tolbutamide, insulin	poorly controlled	50	+	+	n.a.	12 months	n.a.
Case 6, Ellenberg, 1974 [1]	M	60	n.a.	n.a.	tolazamide	well-controlled	53	+	+	n.a.	n.a.	n.a.
Case 1, Chandler, 1978 [9]	M	62	n.a.	11	tolbutamide	well-controlled	56	n.a.	+	n.a.	n.a.	n.a.
Case 1, Gade, 1980 [10]	M	59	n.a.	12	tolbutamide	well-controlled	n.a.	+	+	fluphenazine, amitriptyline	6 months	n.a.
Case 1, Archer, 1983 [11]	M	32	IDDM	8	insulin	n.a.	21	+	+	n.a.	n.a.	n.a.
Case2, Archer, 1983 [11]	M	41	NIDDM	7	chlorpropamide, insulin	n.a.	21	n.a.	+	n.a.	n.a.	n.a.
Case 3, Archer, 1983 [11]	M	13	IDDM	n.a.	insulin	n.a.	21.6	n.a.	n.a.	n.a.	n.a.	n.a.
Case 4, Archer, 1983 [11]	M	61	NIDDM	n.a.	OHA, insulin	n.a.	12	n.a.	n.a.	n.a.	n.a.	n.a.
Case 5, Archer, 1983 [11]	M	59	IDDM	n.a.	insulin	n.a.	16	n.a.	n.a.	n.a.	n.a.	n.a.
Case 6, Archer, 1983 [11]	M	46	NIDDM	n.a.	OHA, insulin	n.a.	35	n.a.	n.a.	n.a.	n.a.	n.a.
Case 7, Archer, 1983 [11]	M	58	NIDDM	n.a.	OHA, insulin	n.a.	12	n.a.	n.a.	n.a.	n.a.	n.a.
Case 8, Archer, 1983 [11]	M	47	IDDM	n.a.	insulin	n.a.	n.a.	n.a.	n.a.	n.a.	n.a.	n.a.
Case 9, Archer, 1983 [11]	M	41	IDDM	12	chlorpropamide, insulin	n.a.	16.5	+	+	analgesic diazepam	n.a.	n.a.
Case 1, Blau, 1983 [12]	F	64	NIDDM	3	chlorpropamide	well-controlled	57	+	///	n.a.	n.a.	n.a.
Case 1, D’Costa, 1992 [13]	M	46	T2DM	3 months	gliclazide, insulin	HbA1c 8%	25	+	n.a.	analgesic TCA	n.a.	n.a.
Case 2, D’Costa, 1992 [13]	M	67	NIDDM	n.a.	OHA, insulin	HbA1c 7.8%	27	+	n.a.	analgesic TCA	n.a.	n.a.
Case 3, D’Costa, 1992 [13]	F	55	NIDDM	n.a.	OHA, insulin	HbA1c 8.2%	17	+	///	analgesic TCA	n.a.	n.a.
Case 4, D’Costa, 1992 [13]	M	53	IDDM	n.a.	insulin	HbA1c 11.7%	18	+	n.a.	analgesic TCA	n.a.	n.a.
Case 1, Godil, 1996 [14]	M	57	NIDDM	n.a.	gliburide	HbA1c 6.9%	n.a.	+	+	amitriptyline	n.a.	n.a.
Case 1, Weintrob, 1997 [15]	F	19	T1DM	9	insulin	n.a.	n.a.	+	///	linolenic acid	n.a.	n.a.
Case 1, Al-Hajeri, 2009 [16]	M	35	T2DM	3	OHA, insulin	HbA1c 11.9%	58%	+	+	SSRI TCA gabapentin	n.a.	n.a.
Case 1, Deorchis, 2013 [17]	F	16	T1DM	2	insulin	HbA1c 16.4%	24.2	+	///	NSAID TCA pregabalin	n.a.	n.a.
Case 1, Datta, 2013 [18]	F	42	T2DM	1	OHA	HbA1c 16.9%	n.a.	+	///	n.a.	n.a.	n.a.
Case 1, Naccache, 2014 [19]	M	71	T2DM	29	insulin, DPP-IVi metformin	HbA1c 9.1%	n.a.	+	n.a.	pregabalin	n.a.	n.a.
Case 1, Gi June Min, 2015 [20]	F	50	T2DM	10	insulin, nateglinide metformin	HbA1c 15%	35	+	///	nortryptiline gabapentin acetaminophentramadol	n.a.	n.a.
Case 1, Iyagba, 2016 [21]	F	40	n.a.	12	insulin	HbA1c 13.2%	n.a.	n.a.	///	carbamazepinepregabalin	n.a.	n.a.
Case 1, Yusof, 2019 [22]	F	34	T2DM	1	metformin, gliclazide	HbA1c 6.2%	38	-	///	amitriptyline	6 months	n.a.

Footnotes: DPP-IVi, Dipeptidyl Peptidase-IV inhibitor; F, female; HbA1c, glycated hemoglobin; IDDM, insulin-dependent diabetes mellitus; M, male; n.a. not available; NIDDM, non-insulin-dependent diabetes mellitus; NSAID, non-steroidal anti-inflammatory drug; OHA, oral hypoglycemic agents; SSRI, selective serotonin reuptake inhibitor; T1DM, type 1 diabetes mellitus; T2DM, type 2 diabetes mellitus; TCAs, tricyclic antidepressant TCA.

## Data Availability

The clinical data presented in this study are available on request from the corresponding author.

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
