# Peer review of "Diabetic Neuropathic Cachexia: A Clinical Case and Review of Literature"

_life, 2022, doi:10.3390/life12050680_

Round 1

Reviewer 1 Report

This paper reports a recent case of diabetic neuropathic cachexia and compare it to previous cases, which is valuable for the exploration of the pathophysiological basis of diabetic neuropathic cachexia.

  1. Section 3 Investigation. Please add the depression screening and list the detailed symptoms.
  2. Line 213-214 “With this in mind, it could be speculated that the appropriate therapeutic choice in diabetic neuropathic cachexia is re-alimentation and recovery of weight.” Please discuss the effect of weigh recovery% on the pain recovery. Is the full weight recovery necessary? Or recovery to the range of normal BMI is efficient for the pain recovery?

Reviewer 2 Report

In this article, A. Belelli et al. describe a case "diabetic neuropathic cachexia" in a patient with type 2 diabetes. In addition, other descriptions of this complication from the literature have been analyzed.

In this case, many questions remain. The development of this form of neuropathy is usually associated with a rapid decrease in glucose levels or long-term hyperglycemia. In the case described, these factors are not obvious: there are no data on previous glycemia and the dynamics of glycemia during hospitalization are not presented. HbA1c concentration does not correspond to the only reported glucose level, even with adjustment to anemia (possible stress hyperglycemia against the background of a perforated ulcer, surgery?). Weight loss in this situation could be caused by poor nutrition caused by depression or gastric ulcer, as well as insulin deficiency. It cannot be ruled out that this is a case of depression, which led to a decrease in nutrition, weight and vitamin B12 deficiency. It is hardly possible to consider this case as cachexia, since the patient's body weight is 75 kg.

Unfortunately, the manuscript adds very little to what is known about the complication. It might be appropriate to focus the discussion on differential diagnosis.

Title: it should be “clinical case’ instead of “clinic case”.

Discussion. Line 153: patients is not young, he is middle-aged.

Round 2

Reviewer 2 Report

The authors revised the manuscript taking into account the recommendations of the reviewers. Some details of the anamnesis have been clarified. Important points in the differential diagnosis were included in the discussion. Controversial aspects of this clinical case are underlined. In my opinion, the revision significantly improved the quality of the manuscript and the potential interest of the audience.